# Inactivation of SARS-CoV-2 and influenza A virus by dry fogging hypochlorous acid solution and hydrogen peroxide solution

**Masahiro Urushidani[1], Akira Kawayoshi[1], Tomohiro Kotaki[2,3], Keiichi Saeki[4], Yasuko Mori[5], Masanori Kameoka**[2] *

**1** Disinfection Division, H. Ikeuchi & Co., Ltd., Nishiwaki, Hyogo, Japan, **2** Division of Global Infectious Diseases, Department of Public Health, Kobe University Graduate School of Health Sciences, Kobe, Hyogo, Japan, **3** Department of Virology, Research Institute for Microbial Diseases, Osaka University, Suita, Osaka, Japan, **4** Laboratory of Microbiology and Immunology, Graduate School of Agricultural Science, Kobe University, Kobe, Hyogo, Japan, **5** Division of Clinical Virology, Center for Infectious Diseases, Kobe University Graduate School of Medicine, Kobe, Hyogo, Japan

* mkameoka@port.kobe-u.ac.jp

## Abstract

Severe acute respiratory syndrome coronavirus 2 (SARS-CoV-2), the causative agent of coronavirus disease 2019 (COVID-19), is transmitted mainly by droplet or aerosol infection; however, it may also be transmitted by contact infection. SARS-CoV-2 that adheres to environmental surfaces remains infectious for several days. We herein attempted to inactivate SARS-CoV-2 and influenza A virus adhering to an environmental surface by dry fogging hypochlorous acid solution and hydrogen peroxide solution. SARS-CoV-2 and influenza virus were air-dried on plastic plates and placed into a test chamber for inactivation by the dry fogging of these disinfectants. The results obtained showed that the dry fogging of hypochlorous acid solution and hydrogen peroxide solution inactivated SARS-CoV-2 and influenza A virus in CT value (the product of the disinfectant concentration and contact time)-dependent manners. SARS-CoV-2 was more resistant to the virucidal effects of aerosolized hypochlorous acid solution and hydrogen peroxide solution than influenza A virus; therefore, higher concentrations of disinfectants or longer contact times were required to inactivate SARS-CoV-2 than influenza A virus. The present results provide important information for the development of a strategy that inactivates SARS-CoV-2 and influenza A virus on environmental surfaces by spatial fogging.

**Data Availability Statement:** All relevant data are within the paper and its Supporting Information files.

## Introduction

Coronavirus disease 2019 (COVID-19) continues to spread worldwide, with more than 424 million individuals being infected and more than 5.88 million dying to date [1]. The World Health Organization (WHO) has recommended a number of countermeasures to the public, including getting vaccinated, avoiding 3Cs (spaces that are closed, crowded, or involve close contact), wearing a properly fitting mask when physical distancing is not possible and in

**Funding:** This work was supported in part by a research fund from H. Ikeuchi & Co. for the industry-academia joint research project (for M.U., A.K., T.K., K.S., Y.M. and M.K.); a research grant by the Kobe University Graduate School of Health Sciences (for M.K.); and a research grant by the Japan Agency for Medical Research and Development (AMED) under grant number JP20nk0101634 (for M.K.). The funders had no role in the study design, data collection and analysis, decision to publish, or preparation of the manuscript. The funder "H. Ikeuchi & Co." provided support in the form of salaries for the authors, Urushidani, M. and Kawayoshi, A., but did not have any additional role in the study design, data collection and analysis, decision to publish, or preparation of the manuscript.

**Competing interests:** The authors have declared that no competing interests exist.

poorly ventilated settings, and frequently cleaning hands with an alcohol-based hand rub or soap and water [2]; however, the pandemic continues.

Severe acute respiratory syndrome coronavirus 2 (SARS-CoV-2) is the causative agent of COVID-19. Viral transmission is established by inhaling droplets or aerosols containing the virus that are excreted from infected individuals in a 3Cs setting or by touching the eyes, nose, or mouth with hands contaminated with the virus adhering to environmental surfaces [3]. Previous studies reported the high stability of SARS-CoV-2 adhering to environmental surfaces [4, 5], and its stability was shown to be higher than those of SARS-CoV and influenza A virus [6, 7]. Therefore, the disinfection of environmental surfaces is indispensable as an infection control measure. However, it is not realistic to frequently and manually disinfect the surfaces of large spaces, such as a train station or airport, because of the manpower and time required. Although the fogging of a disinfectant is an alternative spatial disinfection method, it is not recommended by the WHO due to its effects on the human body [8]. In addition, the Center for Disease Control and Prevention of the United States of America (USA) does not recommend the fogging of disinfectants in hospital rooms in the 2003 and 2008 guidelines [9]. Newer technologies for performing disinfectant fogging were assessed in the 2011 guidelines for the prevention and control of norovirus gastroenteritis outbreaks in healthcare settings; nevertheless, further research is required to clarify the effectiveness and reliability of disinfectant fogging [9].

Therefore, the present study examined the effectiveness of inactivating SARS-CoV-2 virus adhering to plastic microplates by dry fogging disinfectants. Dry fog is defined as an aerosol with a Sauter mean droplet diameter ≤10 μm and maximum droplet diameter ≤50 μm [10], and it does not wet objects even if touched. Its virucidal effects on influenza A virus, which is a common envelope virus transmitted through droplets and contact transmission worldwide, were also investigated. In consideration of the effects of residues after dry fogging on the human body, we tested hypochlorous acid solution and hydrogen peroxide solution, which leave almost no residue on environmental surfaces after dry fogging. Hypochlorous acid solution is an aqueous solution that contains hypochlorous acid (HOCl) as the main component. Hypochlorous acid solution may be prepared by dissolving sodium hypochlorite in water with adjustments to a weak acidic pH. The main component of the weakly acidic solution is HOCl, while that of the alkaline solution is hypochlorite ions (OCl-). HOCl exerts stronger bactericidal effects than OCl- [11]. Therefore, a weakly acidic (pH 6.5) hypochlorous acid solution was dry fogged in the present study.

## Materials and methods

### Preparation of disinfectants

Hypochlorous acid solution and hydrogen peroxide solution were prepared as disinfectants to be dry fogged for the inactivation of viruses. Commercially available, weakly acidic (pH 6.5) hypochlorous acid solution with a free available chlorine (FAC) concentration (the sum of HOCl and OCl- concentrations) [11] of 250 ppm (Super Jiasui; HSP Corporation, Okayama, Japan) and a solution diluted by distilled water with a FAC concentration of 125 ppm were used. To prepare a solution with a FAC concentration of 8,700 ppm, sodium hypochlorite (Hayashi Pure Chemical Ind., Ltd., Osaka, Japan) was dissolved in distilled water, followed by an adjustment of the pH of the solution to 6.5 with HCl (Hayashi Pure Chemical Ind., Ltd.) using the pH meter SK-620PH (Sato Keiryoki Mfg. Co., Ltd, Tokyo, Japan). Commercially available hydrogen peroxide solution (56,400 ppm; Part 1, Decon7; Decon7 Systems LLC., Texas, USA) and solutions diluted by distilled water with hydrogen peroxide concentrations of

11,280, 5,640, 2,820, and 1,410 ppm were prepared. Regarding the negative control, distilled water was dry fogged.

## Cells and viruses

VeroE6/TMPRESS2 cells (JCRB1819) [12] were maintained in Dulbecco's modified Eagle medium (DMEM) (Nacalai Tesque, Inc., Kyoto, Japan) supplemented with 10% fetal bovine serum (Sigma-Aldrich, Merck, Kenilworth, New Jersey, USA) and 1 mg/ml of G418 (Sigma-Aldrich) (complete DMEM), while MDCK cells were maintained in Eagle's minimum essential medium (MEM) (MEM1, Nissui Pharmaceutical Co., Ltd., Tokyo, Japan) supplemented with 10% FBS in a $CO_2$ incubator. The SARS-CoV-2 Wuhan strain, SARS-CoV-2/Hu/DP/Kng/19-020 was propagated by infecting VeroE6/TMPRESS2 cells and cultured in FBS-free DMEM supplemented with 1 mg/ml of G418 for 24 hours. The influenza A H1N1 strain, A/Puerto Rico/8/1934 was propagated by infecting MDCK cells and cultured in FBS-free MEM supplemented with 2 μg/ml of acetylated trypsin (Sigma-Aldrich) for 3 days. Viral supernatants were clarified by centrifugation, aliquoted, and stored at -85°C. Ten-fold serially diluted viral supernatants were incubated with VeroE6/TMPRESS2 or MDCK cells for 3 or 4 days, respectively, for viral titration, and median tissue culture infectious doses ($TCID_{50}$) were measured using the Spearman-Kaber method following the fixation of cells with 5% formaldehyde in phosphate-buffered saline (PBS) and staining with 0.5% crystal violet in 20% EtOH. $TCID_{50}$ values below the detection limit ($3.16 \times 10^2$ $TCID_{50}$/ml or 2.5 log10 $TCID_{50}$/ml) were assigned to half of the detection limit, equivalent to $1.58 \times 10^2$ $TCID_{50}$/ml or 2.2 log10 $TCID_{50}$/ml, because substituting the value to half of the detection limit was previously shown to be less biased than substitution to zero or the detection limit [13].

## Preparation of air-dried virus samples

Viral solutions (5 μl) containing SARS-CoV-2 ($1.2 \times 10^5$ $TCID_{50}$) or influenza A virus ($2.8 \times 10^6$ $TCID_{50}$) were applied using a micropipette to the bottom of 5 wells (per virus inactivation experiment) on a 96-well flat-bottomed microplate (Corning Japan, Shizuoka, Japan), air-dried for 10–15 minutes using a small electric fan, and subjected to a virus inactivation experiment. As a negative control for the experiment, an air-dried viral sample in a well was re-suspended with 200 μl of DMEM containing a neutralizer of the disinfectant prior to dry fogging, as described below. In addition, in some experiments, 5 μl of artificial saliva (Saliveht aerosol; Teijin Pharma, Tokyo, Japan) or PBS was mixed with 5 μl of viral supernatants prior to air-drying samples.

## Preparation of the test chamber for dry fogging

A closed test chamber was prepared to fill the space with dry fog. The detailed settings of the chamber are shown in Fig 1. The chamber size was $500 \times 700 \times 300$ mm (height × width × depth), made of acrylic, and set in a biosafety cabinet of class II type A/B3 or class II type A1. A sliding door was set on the front of the chamber for the handling of samples inside the chamber. A fogger equipped with an impinging-jet atomizing nozzle [14] [AE-1 (03C), AKI-Mist®"E"; H. Ikeuchi & Co., Ltd., Osaka, Japan] was used to fog disinfectants into the space. To generate aerosolized disinfectants in the form of dry fog, 0.3 MPa compressed air was supplied to the fogger from a compressor (0.2LE-8SB0; Hitachi Industrial Equipment Systems Co., Ltd., Tokyo, Japan). The fogging capacity was 2.3 liters per hour and the Sauter mean droplet diameter was 7.5 μm. In the virus inactivation experiment using dry fogging, four 90-mm Petri dishes containing 20 ml of distilled water, a temperature and humidity sensor (Model RHT-3 Temperature and Humidity Sensor; Sensatec Co., Ltd., Kyoto, Japan), and a

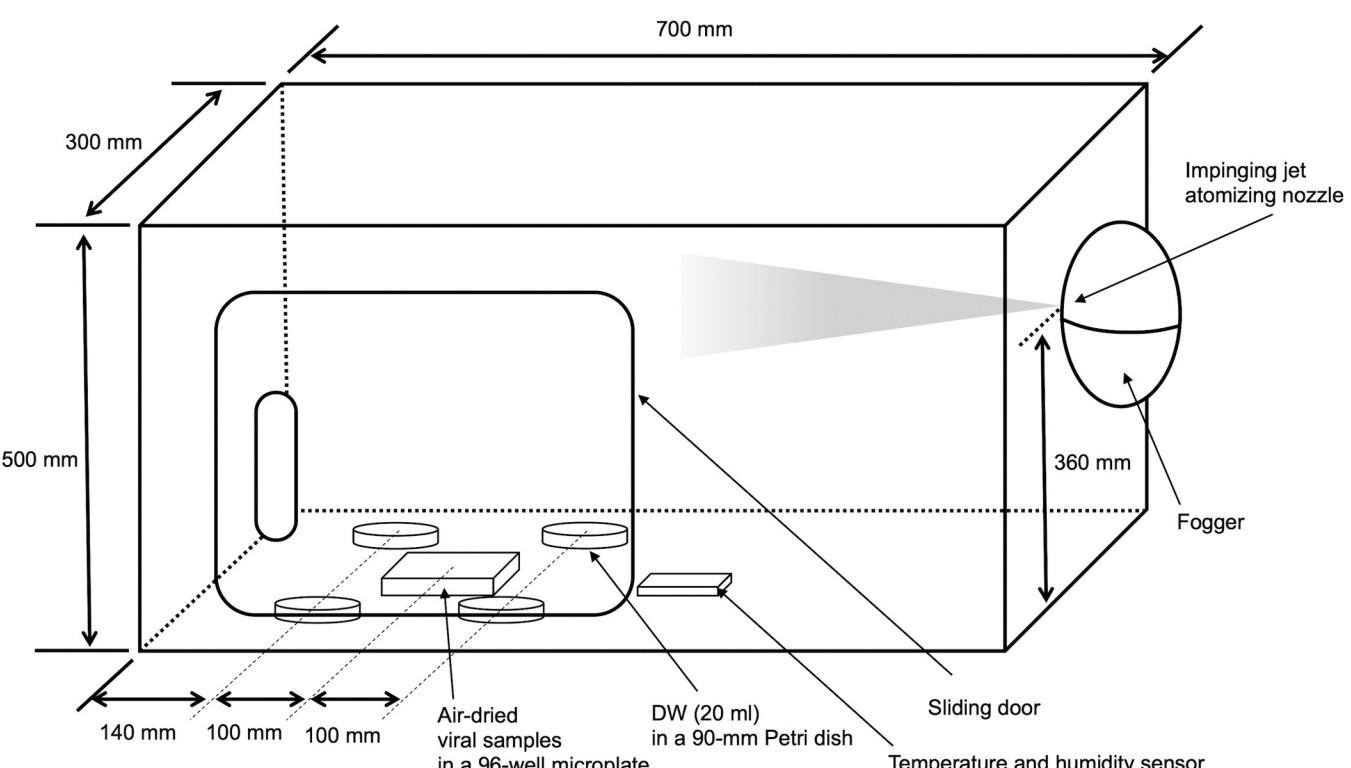

**Fig 1. Schematic illustration of the test chamber.** The composition of the chamber is described in detail in the Materials and methods.

96-well microplate containing air-dried viral samples in 5 wells were placed in the chamber (Fig 1). Four 90-mm Petri dishes containing distilled water were placed to measure the concentrations of disinfectants trapped in water during the virus inactivation experiment and also to calculate the product of the concentrations of disinfectants and contact times (CT value).

## Virus inactivation experiment by dry fogging and measurements of concentrations of disinfectants trapped in distilled water

The times at which spatial fogging, the termination of the virus inactivation operation, and measurements of the concentrations of disinfectants trapped in 20 ml of distilled water were performed as indicated in Fig 2. Spatial fogging was conducted as follows. A disinfectant was dry fogged for 5 seconds at the start of the experiment and left to stand for 4 minutes. Dry fogging was then repeated 3 more times for 2.5 seconds each and left to stand for 4 minutes after each fogging. Dry fogging was performed 4 times, namely, 0, 4, 8, and 12 minutes after the initiation of the experiment, and the total experimental period was 16 minutes. This dry fogging operation allowed for the continuous generation of dry fog without unnecessarily moistening the environmental surface in the chamber, and dry fog did not fade. The virus inactivation reaction by dry fogging disinfectants was terminated by resuspending air-dried viral samples in 200 μl of DMEM containing a neutralizer of the disinfectant being tested. DMEM containing 0.1 M sodium thiosulfate (Hayashi Pure Chemical Ind., Ltd.) was used as the neutralizer for hypochlorous acid solution, while DMEM containing 0.1 mg/ml of catalase (Nacalai Tesque, Inc.) was used as that for hydrogen peroxide solution. These neutralizers did not affect cell growth under experimental conditions (data not shown). The residual infectious titer of viral samples after the inactivation experiment was evaluated by measuring the $TCID_{50}$ value,

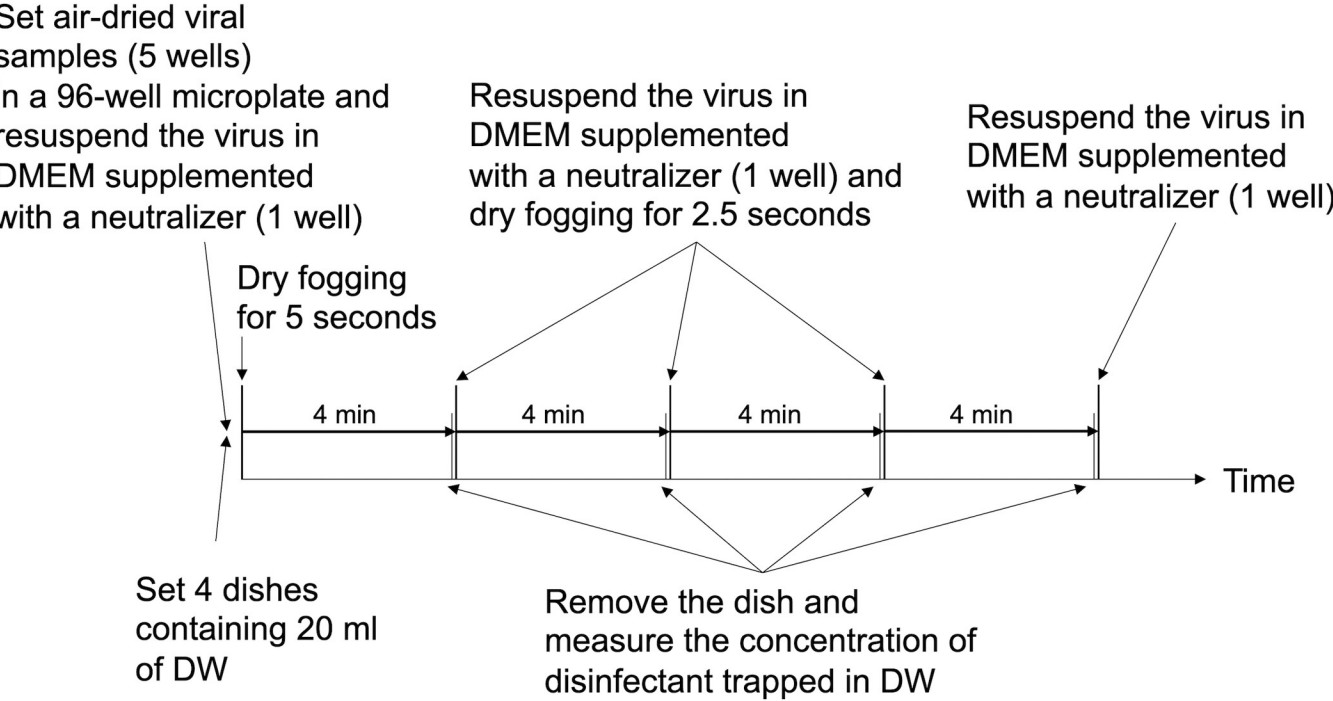

**Fig 2. Flow diagram of the virus inactivation experiment.** A detailed procedure is described in the Materials and methods.

as described above. The concentration of the dry fogged disinfectant was assessed by collecting the droplet-shaped disinfectant that had fallen into 20 ml of distilled water in the 90-mm Petri dish placed in the chamber, and measuring the concentration of the disinfectant dissolved in distilled water. The FAC concentration when hypochlorous acid solution was dry fogged was measured by DPD (N,N-dimethyl-p-phenylenediamine) absorptiometry using a residual chlorine meter (HI96771; Hanna Instruments, Chiba, Japan). The hydrogen peroxide concentration when hydrogen peroxide solution was dry fogged was measured by 4-aminoantipyrine absorptiometry using an enzyme with a hydrogen peroxide concentration meter (DPM2-$H_2O_2$; Kyoritsu Chemical-Check Lab., Corp., Kanagawa, Japan). Concentrations in appropriately diluted samples with distilled water were within the detection ranges of the measuring instruments and reagents. In addition, air temperature and humidity in the chamber were monitored during experiments.

A set of viral inactivation experiments was performed as follows. Four 90-mm Petri dishes containing distilled water, a temperature and humidity sensor, and a 96-well microplate containing air-dried viral samples in 5 wells were placed in the test chamber (Fig 1), as described above. Prior to the dry fogging of disinfectants, a dried viral sample in the first well was resuspended with DMEM (200 µl) containing a neutralizer by pipetting. The sliding door was then closed, and the first dry fogging of the disinfectant was performed for 5 seconds. After 4 minutes, the sliding door of the chamber was opened, the dried viral sample in the second well was resuspended with DMEM (200 µl) containing the neutralizer, and the first Petri dish containing distilled water was removed from the chamber. The sliding door was closed, and the second dry fogging of the disinfectant was performed for 2.5 seconds. After 4 minutes, the sliding door was opened, the dried viral sample in the third well was resuspended with DMEM (200 µl) containing the neutralizer, and the second Petri dish containing distilled water was removed from the chamber. The sliding door was again closed, and the third dry fogging of

the disinfectant was performed for 2.5 seconds. After 4 minutes, the sliding door was opened, the dried viral sample in the fourth well was resuspended with DMEM (200 μl) containing the neutralizer, and the third Petri dish containing distilled water was removed from the chamber. The sliding door was closed, and the fourth dry fogging of the disinfectant was performed for 2.5 seconds. After 4 minutes, the sliding door was opened, the dried viral sample in the fifth well was resuspended with DMEM (200 μl) containing the neutralizer, and the fourth Petri dish containing distilled water was removed from the chamber. The concentrations of the droplet-shaped disinfectant that fell into distilled water in the Petri dishes were measured between dry fogging. Therefore, the concentrations of the disinfectant trapped in distilled water were measured 4 times in each set of experiments, i.e., at 4 minutes (total of 5 seconds of dry fogging), 8 minutes (total of 7.5 seconds of dry fogging), 12 minutes (total of 10 seconds of dry fogging), and 16 minutes (total of 12.5 seconds of dry fogging). Virus inactivation experiments under the same conditions were repeated more than three times, except for those to evaluate artificial saliva, which were repeated twice.

## Statistical analysis

Statistical analyses were performed using the standard function of GraphPad Prism 8 software (GraphPad Software, San Diego, California) with a 2-way ANOVA, unpaired *t*-test, non-linear regression (curve fit) analysis, or simple linear regression analysis.

## Results

### The dry fogging of hypochlorous acid solution and hydrogen peroxide solution inactivated SARS-CoV-2 and influenza A virus over time

We examined changes in the viral infectious titer upon dry fogging with various concentrations of hypochlorous acid solution (FAC concentrations of 8,700, 250, and 125 ppm), hydrogen peroxide solution (56,400, 11,280, 5640, 2820, and 1410 ppm of hydrogen peroxide), or distilled water. Dry fogging experiments were initially conducted using a commercially available hypochlorous acid solution (250 ppm; Super Jiasui; HSP); however, SARS-CoV-2 was not inactivated (Fig 3A). A previous study reported that when viral culture fluid was mixed with 35 ppm hypochlorous acid solution, SARS-CoV-2 was effectively inactivated [15]; therefore, we calculated the FAC concentration needed to inactivate an air-dried virus that settled in the wells (0.32 cm$^2$) of 96-well microplates. The result obtained revealed that 8,700 ppm hypochlorous acid solution in the form of dry fog was required to inactivate SARS-CoV-2 under our experimental conditions. Fig 3 shows the time course of changes in virus infectious titers under the experimental conditions employed in the present study. The viral titer of SARS-CoV-2 was not reduced by 250 ppm hypochlorous acid solution (Fig 3A), whereas that of influenza A virus was effectively decreased over time (Fig 3B). Furthermore, 8,700 ppm hypochlorous acid solution effectively reduced the infectious titer of SARS-CoV-2 (Fig 3A). Moreover, 56,400 ppm hydrogen peroxide solution reduced the infectious titer of SARS-CoV-2 (Fig 3C), while 11,280 ppm hydrogen peroxide solution decreased the viral titer of influenza A virus (Fig 3D). It is important to note that since dry fogging was performed 4 times at 4-minute intervals in the virus inactivation experiment, the concentration of the disinfectant increased over time. Nevertheless, viruses were inactivated over time under the experimental conditions used. In contrast, the dry fogging of distilled water did not reduce the viral infectivity of SARS-CoV-2 or influenza A virus regardless of the elapsed time (Fig 3).

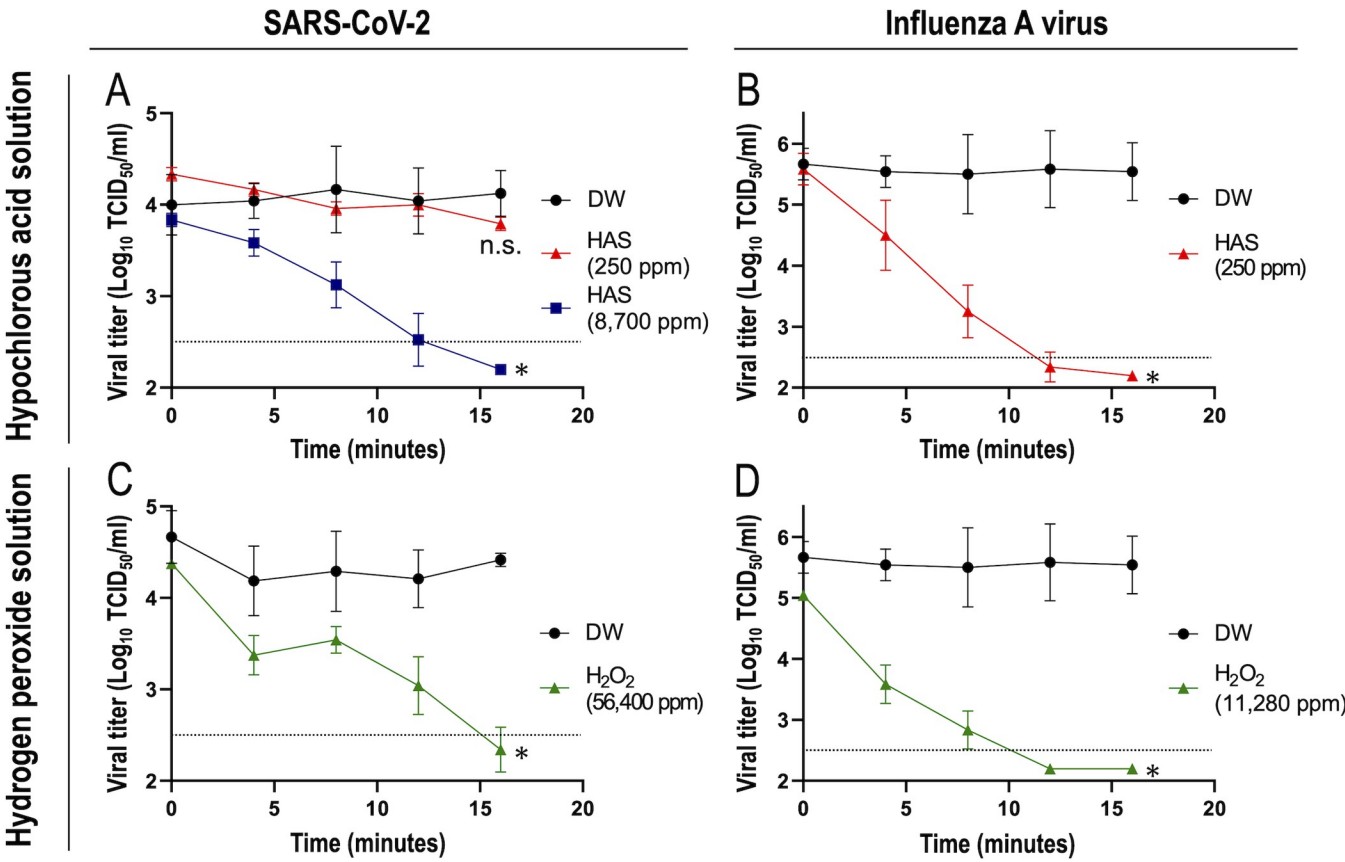

**Fig 3. Inactivation of viruses by the dry fogging of disinfectants.** Changes in the viral infectious titers (TCID$_{50}$ values) of SARS-CoV-2 (A and C) and influenza A virus (B and D) upon dry fogging with hypochlorous acid solution (HAS) (A and B), hydrogen peroxide solution (H$_2$O$_2$) (C and D), or distilled water (DW) (A, B, C, and D) were evaluated, as described in the Materials and methods. Horizontal dotted lines in the graphs show the detection limit of the viral titer. *P < 0.0001 compared with DW using a two-way ANOVA. n.s., not significant. Each data point represents the average and standard deviation obtained from more than three repeated experiments.

### Relationship between dry fogged disinfectant concentrations and concentrations of disinfectants trapped in distilled water

To calculate the CT value of a disinfectant, we measured its dissolved concentration in 20 ml of distilled water in 90-mm Petri dishes during virus inactivation experiments. Raw data on the concentrations of disinfectants that dissolved in distilled water are shown in S1 Table. Regarding the dry fogging of hypochlorous acid solution (125, 250, and 8,700 ppm) and hydrogen peroxide solution (1,410, 2,820, 5,640, 11,280, and 56,400 ppm), the total fogging time versus the concentrations of disinfectants trapped in distilled water as well as the concentrations of dry fogged disinfectants versus the concentrations of disinfectants trapped in distilled water per unit fogging time are plotted in Fig 4.

The results obtained confirmed that the dry fogging of various concentrations of hypochlorous acid solution and hydrogen peroxide solution increased their concentrations in distilled water in proportion to the total fogging time (Fig 4A and 4B). In addition, the concentrations of disinfectants in distilled water per unit fogging time increased in proportion to the dry fogged disinfectant concentration of each disinfectant (Fig 4C and 4D). On one hand, these were expected results because the amount of the droplets of disinfectants from the fogger was proportional to the total fogging time, and the concentrations of disinfectants contained in the

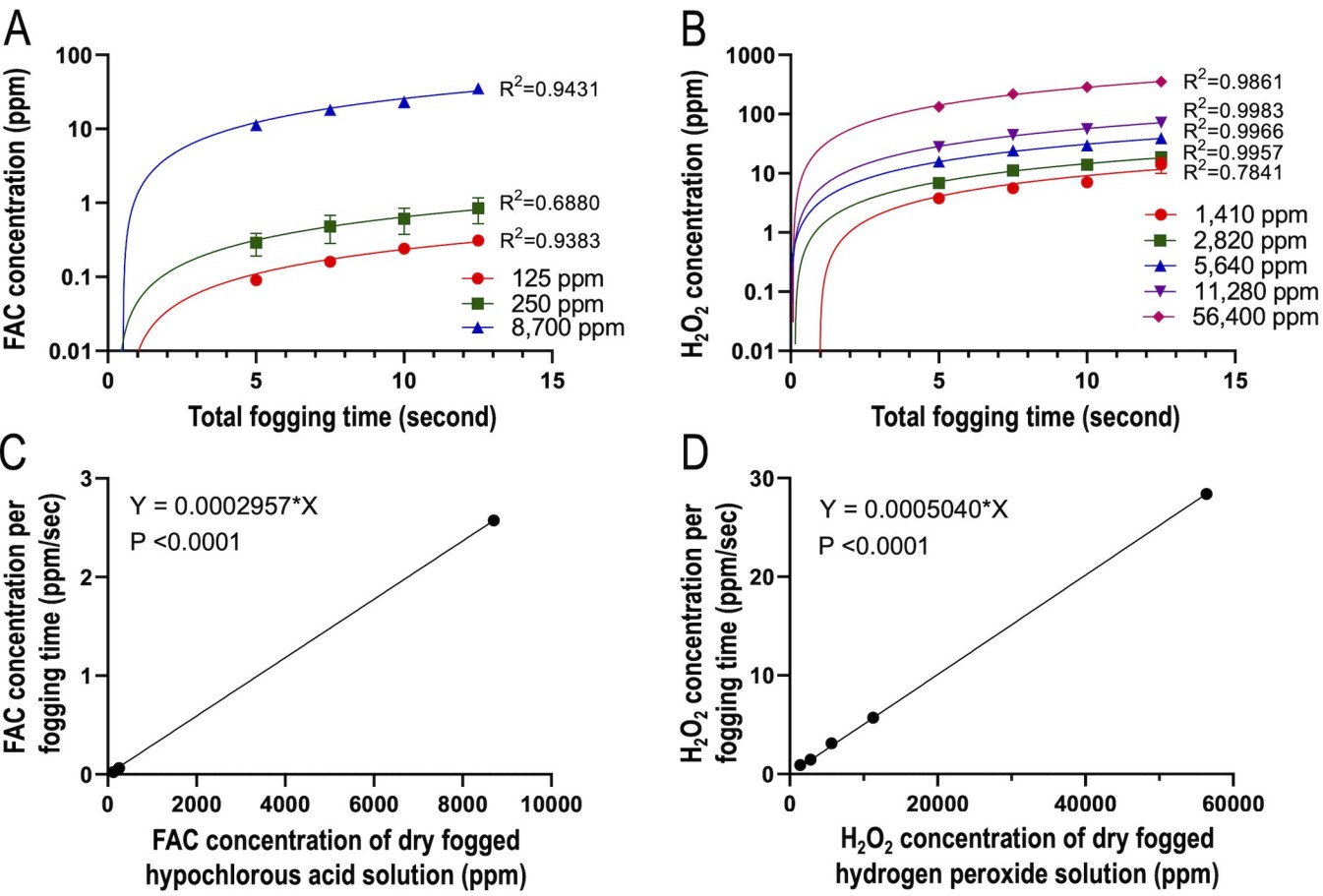

**Fig 4. Relationship between dry fogged disinfectant concentrations and concentrations of disinfectants trapped in 20 ml of distilled water in Petri dishes.** The dry fogging of hypochlorous acid solution (A) or hydrogen peroxide solution (B) was performed as described in the Materials and methods. The concentrations of free available chlorine FAC (C) and hydrogen peroxide ($H_2O_2$) per unit fogging time (D) were calculated and plotted. R squared ($R_2$) values were estimated using a non-linear regression (curve fit) analysis and reported on the graphs (A and B). In addition, $R_2$ values, equations, and p values were estimated using a simple linear regression analysis and reported on the graphs (C and D).

droplets was proportional to disinfectant concentrations. On the other hand, the slope of hypochlorous acid solution was approximately 0.59-fold smaller than that of hydrogen peroxide solution (Fig 4C and 4D). Since the fogging time was the same for hypochlorous acid solution and hydrogen peroxide solution, the number of droplets that fell and dissolved into distilled water in Petri dishes was equivalent. Therefore, the data obtained indicated that the concentration of FAC in droplets decreased more than that of hydrogen peroxide when they travelled in the form of a dry fog. The temperature inside the chamber was maintained at approximately 19–26˚C, while humidity was maintained at 59–99% during dry fogging experiments.

## Relationship between virucidal effects of dry fogged disinfectants and the CT value

The model of Eq (1) calculated from Chick Watson's law is often used to show the bactericidal effects of a disinfectant on microorganisms [11].

$$\log (N/N0) = -kCT \tag{1}$$

In the equation, N0 is the initial bacterial count, N is the viable bacterial count at the contact time (T) of bacteria to the disinfectant, C is the disinfectant concentration, and k is the inactivation rate constant of the bacteria. In the present study, we evaluated the virucidal effects of dry fogged disinfectants by CT values, the product of disinfectant concentrations and contact times, similar to the bacteria inactivation experiment. The concentration of the disinfectant trapped in 20 ml of distilled water in Petri dishes was used as the concentration (C). Fig 5 shows the logarithmic values of viral infectious titers at various CT values. The viral titers of SARS-CoV-2 and influenza A virus linearly decreased with increases in CT values in the dry fogging of hypochlorite acid solution. With the dry fogging of hydrogen peroxide solution, viral titers also linearly decreased. It is important to note that since the logarithmic values of viral infectious titers at various CT values were plotted and fit by a simple linear regression analysis, the solid lines in Fig 5 are shown by Eq (2), which is a modification of Eq (1).

$$\log(I) = -kCT + \log(I0) \qquad (2)$$

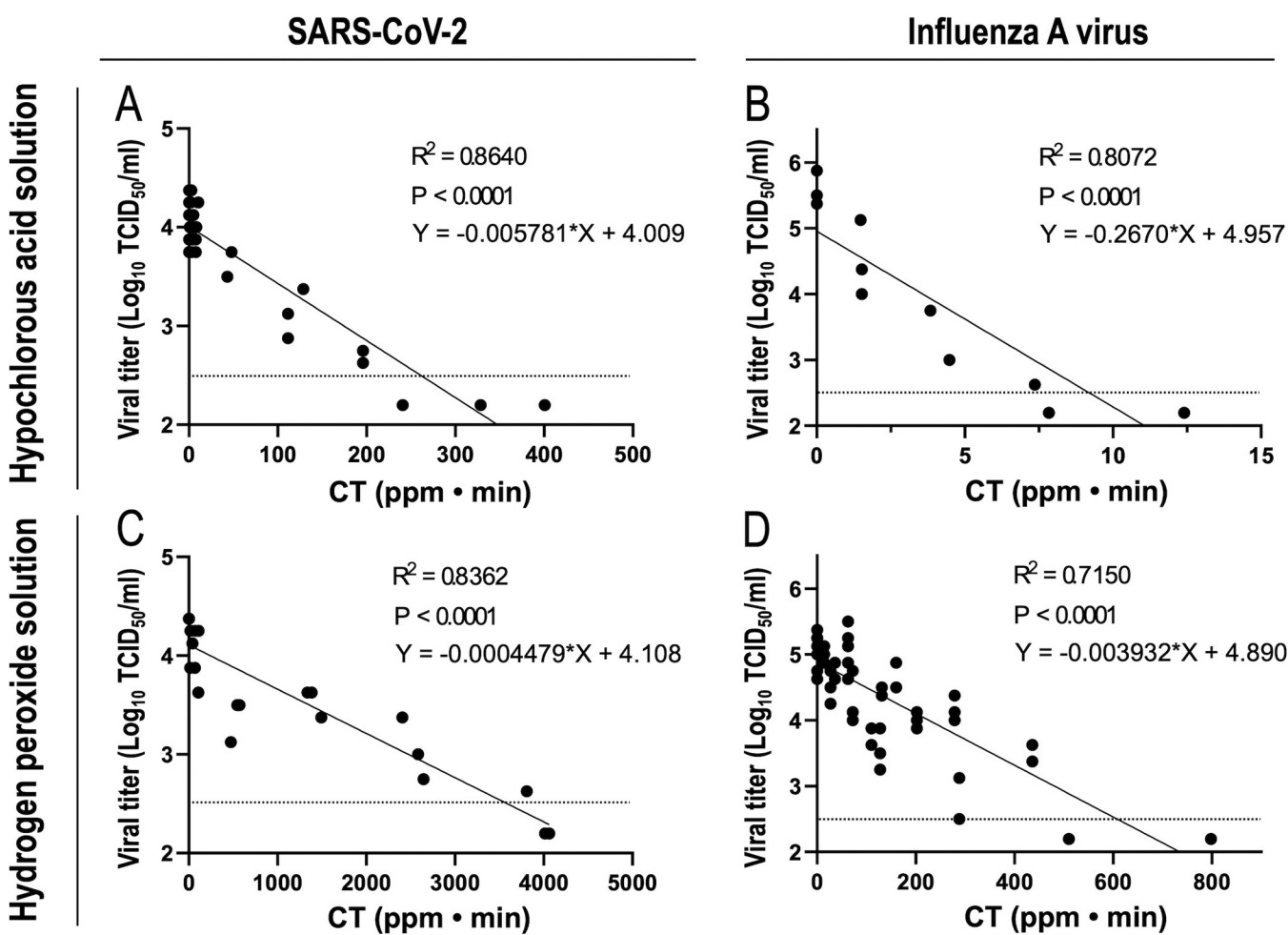

**Fig 5. The relationship between virucidal effects of a disinfectant and the CT value.** Changes in the viral infectious titers (TCID$_{50}$ values) of SARS-CoV-2 (A and C) and influenza A virus (B and D) upon dry fogging with hypochlorous acid solution (A and B) or hydrogen peroxide solution (C and D) were evaluated, as described in the Materials and methods. The CT value of a dry fogged disinfectant was calculated, and a scatter plot was created from each dataset of the viral infectious titer and CT value for the combination of the virus and disinfectant. R squared (R$_2$) values, equations, and p values were estimated using a simple linear regression analysis and reported on the graphs. Horizontal dotted lines in the graphs show the detection limit of the viral titer.

In the equation, $I_0$ is the viral titer before dry fogging, I is the viral titer at the contact time (T) of the virus to a dry fogged disinfectant, and C is the concentration of a disinfectant trapped in 20 ml of distilled water.

In formula (2), k is the inactivation rate constant of the virus. With the dry fogging of hypochlorous acid solution, the inactivation constant of SARS-CoV-2 was approximately 46-fold smaller than that of influenza A virus. In addition, with the dry fogging of hydrogen peroxide solution, the inactivation constant of SARS-CoV-2 was approximately 8.8-fold smaller than that of influenza A virus. Therefore, SARS-CoV-2 was more resistant to hypochlorous acid solution and hydrogen peroxide solution than influenza A virus.

### Effects of saliva on the inactivation of SARS-CoV-2 and influenza A virus by the dry fogging of hypochlorous acid solution and hydrogen peroxide solution

As shown in Figs 3 and 5, the dry fogging of hypochlorous acid solution and hydrogen peroxide solution inactivated SARS-CoV-2 and influenza A virus. We prepared air-dried viral samples using viral supernatants for these experiments, and viral solutions differed from body fluids. Therefore, to assess the effects of saliva components on the virucidal effects of dry fogged disinfectants, air-dried viral samples were prepared by mixing viral supernatants and artificial saliva solution or PBS as the negative control. Hypochlorous acid solution and hydrogen peroxide solution were then dry fogged for viral samples. The results obtained revealed no significant differences in the levels of virus inactivation upon the dry fogging of disinfectants between air-dried viral samples prepared with artificial saliva and PBS (Fig 6).

## Discussion

The inactivation of pathogenic viruses, including SARS-CoV-2, influenza A virus, norovirus, and adenovirus, by the dry fogging of a mixture of peracetic acid and hydrogen peroxide was previously demonstrated [16–21]. Therefore, the dry fogging of disinfectants is considered to effectively inactivate pathogenic viruses on environmental surfaces in laboratories, safety cabinets, and health care facilities. In this report, we investigated whether SARS-CoV-2 and influenza A virus that had been air-dried and adhered to an environmental surface were inactivated by the dry fogging of hypochlorous acid solution or hydrogen peroxide. Hypochlorous acid is decomposed into hydrochloric acid and oxygen, while hydrogen peroxide is decomposed into water and oxygen with time. Since decomposition products other than water are gases at a normal temperature and pressure, they do not remain on environmental surfaces after dry fogging with appropriate ventilation. To the best of our knowledge, the inactivation of SARS-CoV-2 by the dry fogging of hypochlorous acid solution or hydrogen peroxide has not yet been reported. The present results revealed that even though the concentration of the disinfectant required for virus inactivation by dry fogging differed, the infectivities of SARS-CoV-2 and influenza A virus were both reduced to below the detection limit over time with the dry fogging of disinfectants (Fig 3). A previous study reported the higher stability of SARS-CoV-2 than influenza A virus on environmental surfaces [6]. We speculated that the higher stability of SARS-CoV-2 than influenza A virus may be related to our results showing that with dry fogging for the same duration, higher concentrations of hypochlorous acid solution and hydrogen peroxide solution were required to inactivate SARS-CoV-2 than influenza A virus. SARS-CoV-2 and influenza A viruses are both enveloped RNA viruses; however, experiments to elucidate differences in their resistance to the dry fogging of these disinfectants were not conducted in the present study.

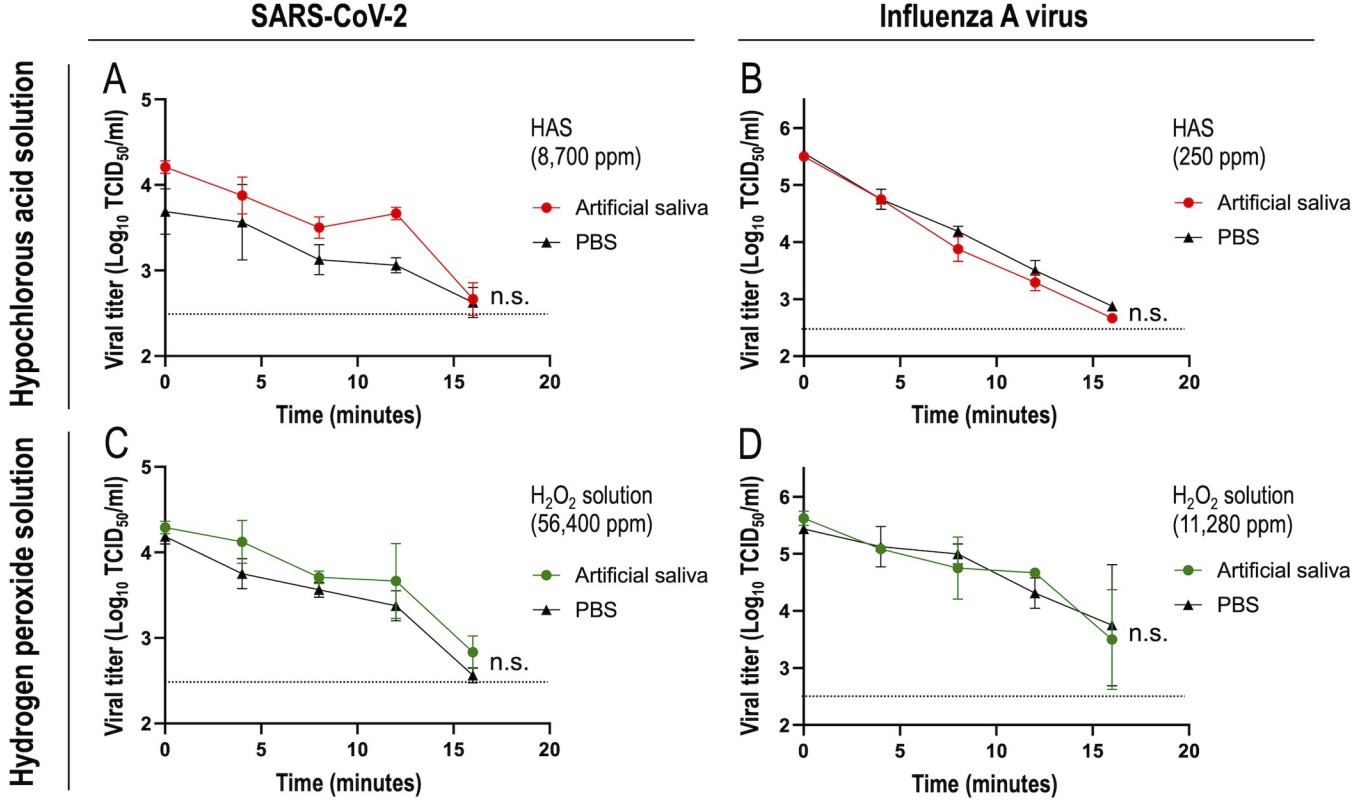

**Fig 6. Effects of saliva on the inactivation of SARS-CoV-2 and influenza A virus by the dry fogging of disinfectants.** SARS-CoV-2 (A and C) or influenza A virus (B and D) was mixed with artificial saliva or PBS, and air-dried. Changes in the viral infectious titers (TCID$_{50}$ values) upon dry fogging with hypochlorous acid solution (HAS) (A and B) or hydrogen peroxide solution were then evaluated, as described in the Materials and methods. Horizontal dotted lines in the graphs show the detection limit of the viral titer. n.s., not significant between artificial saliva- and PBS-containing samples using an unpaired *t*-test. Each data point represents the average and standard deviation obtained from two repeated experiments.

The merits of dry fogging are 1) it does not wet the environmental surface and 2) it is diffusible and has a small droplet size; therefore, an object will not be excessively wet unless fogging is performed in the same place for a long time. With dry fogging, it is not necessary to wipe off the disinfectant after fogging, and it is also possible to use it in an environment that cannot become wet. Regarding diffusivity, since the droplet size is small, droplets are assumed to fall at a low speed and float in the air for a long time. By utilizing this property of dry fog, it is also possible to diffuse droplets using a fan. Furthermore, droplets reach the backs of objects, such as desks and chairs, as well as gaps that cannot be accessed; therefore, dry fogging is considered to be suitable for spatial fogging. On the other hand, one of the disadvantages of dry fogging is the risk of the inhalation of droplets. Therefore, several factors need to be considered, such as the concentration of droplets in space, the staying time, the amount of air being inhaled by the number of breaths, the concentration of the solution in droplets, and the toxicity of the chemical to the human body. In the present study, we assumed an unmanned space and, thus, did not consider risks to the human body; however, when fogging a disinfectant in the form of dry fog in a manned environment, it is extremely important to consider risks to the human body. Therefore, if someone has to be present, they need to be wearing appropriate personal protective equipment and respiratory protection.

The results of the dry fogging of hypochlorous acid solution and hydrogen peroxide solution using the test chamber confirmed that the concentrations of FAC and hydrogen peroxide

trapped in distilled water increased according to the total fogging time and fogged disinfectant concentration (Fig 4). Furthermore, SARS-CoV-2 and influenza A virus were inactivated in a manner that was dependent on CT values (Fig 5). However, the present results were obtained from a fogging experiment conducted in a small space (approximately 0.11 m$^3$), and, thus, various factors need to be considered when dry fogging in actual spaces, such as the vaporization of droplets and reductions in disinfectant concentrations in droplets due to the longer distance travelled by droplets [22, 23]. In the present study, a maximum of approximately 8 ml of disinfectant was dry fogged in a space of approximately 0.11 m$^3$, and it is inadequate to simply calculate the amount of disinfectant needed for the volume of the space to be fogged. It may be necessary to fog for a longer duration or with higher concentrations of a disinfectant in actual spaces. Since the above factors of fogging in actual spaces reflects the concentration of a dry fogged disinfectant, it is important to measure it at the site of use in order to confirm its effectiveness.

Three important factors for the occurrence of infectious diseases are infection sources, transmission routes, and susceptible hosts; therefore, countermeasures need to be taken against these factors. Among them, as a countermeasure against infection sources, patient isolation and quarantine are currently being performed for COVID-19. Regarding more active countermeasures against infection sources, the inactivation of SARS-CoV-2 by UV or ozone irradiation has been examined at basic research levels as part of the attempt to inactivate the virus adhering to environmental surfaces [24–26]. The inactivation of influenza A virus adhering to environmental surfaces using an ultrasonic atomizer of hypochlorous acid solution has been reported [27]. In the present study, we revealed that the dry fogging of hypochlorous acid solution and hydrogen peroxide solution effectively inactivated SARS-CoV-2 and influenza A virus that had been adhered to plastic microplates, suggesting that it is an active countermeasure against infection sources on environmental surfaces.

There are a number of limitations that need to be addressed. Only the SARS-CoV-2 Wuhan strain and influenza A virus H1N1 strain were tested in the present study. SARS-CoV-2 variants of concern, including the delta variant, are continually emerging. Furthermore, there is a wide variety of influenza A viral strains. Inactivation experiments were not performed on these viral strains in the present study. However, a previous study suggested that hypochlorous acid attacks multiple components of microorganisms, including the plasma membrane and nucleic acids, as its germicidal effect [11]. Although the structures of viral proteins due to genetic mutations may differ between the Wuhan strain and other SARS-CoV-2 variants, there may be a commonality in the basic structures and components of virions, such as the lipid bilayer (envelope) and structural and non-structural proteins. A lipid bilayer is the least resistant component of enveloped viruses to disinfection [9]; therefore, dry fogging is expected to act effectively not only against the Wuhan strain and H1N1 strain tested in the present study, but also against other virus strains. As another limitation, we only examined the effects of dry fogging on viruses dried on the surfaces of plastic microplates. Further studies are warranted to investigate its effects on viruses adhering to the surfaces of other materials. Furthermore, it is important to examine whether dry fogging of disinfectants inactivates viruses in a space. Nevertheless, we consider spatial fogging to be an effective method for inactivating SARS-CoV-2 and influenza A virus on environmental surfaces. The accumulation of more information in the future and the development of methods that inactivate pathogens, such as viruses, in the environment for practical use are desired.

## Supporting information

**S1 Table. FAC and $H_2O_2$ concentrations in 20 ml of distilled water in Petri dishes.** (XLSX)

## Acknowledgments

We thank Kenji Ayagi, Issei Toyoda, and Masataka Ishida of the Innovation Commercialization Division, Kobe University for their efforts to coordinate the industry-academia joint research project. The manuscript was proofread by Medical English Service, Kyoto, Japan.

## Author Contributions

**Conceptualization:** Masahiro Urushidani, Akira Kawayoshi, Keiichi Saeki, Yasuko Mori, Masanori Kameoka.

**Funding acquisition:** Masanori Kameoka.

**Investigation:** Masahiro Urushidani, Akira Kawayoshi, Tomohiro Kotaki, Keiichi Saeki, Masanori Kameoka.

**Methodology:** Masahiro Urushidani, Akira Kawayoshi, Tomohiro Kotaki, Keiichi Saeki, Yasuko Mori, Masanori Kameoka.

**Project administration:** Masanori Kameoka.

**Supervision:** Akira Kawayoshi, Masanori Kameoka.

**Validation:** Masanori Kameoka.

**Writing – original draft:** Masahiro Urushidani, Masanori Kameoka.

**Writing – review & editing:** Akira Kawayoshi, Tomohiro Kotaki, Keiichi Saeki, Yasuko Mori.

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
