## [Decision Letter · Decision Letter 0]

12 Jan 2022

PONE-D-21-38754Inactivation of SARS-CoV-2 and influenza A virus by spraying hypochlorous acid solution and hydrogen peroxide solution in the form of Dry FogPLOS ONE

Dear Dr. Kameoka,

Thank you for submitting your manuscript to PLOS ONE. After careful consideration, we feel that it has merit but does not fully meet PLOS ONE’s publication criteria as it currently stands. Therefore, we invite you to submit a revised version of the manuscript that addresses the points raised during the review process. The methodology is important for the criterion of PLOS ONE. Please pay a special attention to the methods.

We look forward to receiving your revised manuscript.

Kind regards,

Etsuro Ito

Academic Editor

PLOS ONE

Journal Requirements:

"This work was supported in part by a research fund from H. Ikeuchi & Co. for the industry-academia joint research project (for M.U., A.K., T.K., K.S., Y.M. and M.K.); a research grant by the Kobe University Graduate School of Health Sciences (for M.K.); and a research grant by the Japan Agency for Medical Research and Development (AMED) under grant number JP20nk0101634 (for M.K.). The funders had no role in the study design, data collection and analysis, decision to publish, or preparation of the manuscript."

We note that one or more of the authors is affiliated with the funding organization, indicating the funder may have had some role in the design, data collection, analysis or preparation of your manuscript for publication; in other words, the funder played an indirect role through the participation of the co-authors. If the funding organization did not play a role in the study design, data collection and analysis, decision to publish, or preparation of the manuscript and only provided financial support in the form of authors' salaries and/or research materials, please do the following:

a. Review your statements relating to the author contributions, and ensure you have specifically and accurately indicated the role(s) that these authors had in your study. These amendments should be made in the online form.

b. Confirm in your cover letter that you agree with the following statement, and we will change the online submission form on your behalf: 

“The funder provided support in the form of salaries for authors [insert relevant initials], but did not have any additional role in the study design, data collection and analysis, decision to publish, or preparation of the manuscript. The specific roles of these authors are articulated in the ‘author contributions’ section.

Reviewers' comments:

Reviewer's Responses to Questions

**Comments to the Author**

1. Is the manuscript technically sound, and do the data support the conclusions?

Reviewer #1: Partly

Reviewer #2: Yes

2. Has the statistical analysis been performed appropriately and rigorously? 

Reviewer #1: I Don't Know

Reviewer #2: Yes

3. Have the authors made all data underlying the findings in their manuscript fully available?

Reviewer #1: No

Reviewer #2: Yes

4. Is the manuscript presented in an intelligible fashion and written in standard English?

Reviewer #1: No

Reviewer #2: Yes

5. Review Comments to the Author

Reviewer #1: In title and throughout paper, the term “dry fog” does not need to be capitalized, since it is not a proper noun. this may be an issue with English

Line 21: you should clarify that the current thinking is that the predominant mechanism for transmission is via air (small particles, up to droplets), but that contact with contaminated surfaces may be a possibility.

57: confusing “spraying” for “fogging” they are not the same.

59-63: the sentence is a bit long, and somewhat inaccurate. Check the reference again, but after looking at it myself, it seems the CDC is ok with spraying, and using some types of foggers. Maybe the authors are confusing fogging with spraying, due to English being their second language.

65: change “spraying” to “dispensing”… this comment applies throughout the paper.

67: add reference for your dry fog definition.

85: indicate how pH was measured

118: suggest starting a new paragraph here in discussing inoculation of the viruses into the well plates. Also need to add some more details about this. For example, need to address how were the viruses added, how many well plates were inoculated with virus each experiment, how the positive controls were handled, etc.

130: so a sprayer was used and not a fogger? It’s confusing.

136: explain what the 4 petri plates of water were used for

162: what is DPD? Confirm that the chlorine meter measures free available chlorine and not just other species of chlorine

169: indicate what was used to measure temp and relative humidity

170-180: the paragraph about how the experiments were conducted is not very clear.

171: need to clarify how many viral samples were resuspended with the DMEM/neutralizer. I’m assuming maybe ¼ of samples were treated this way… and then another ¼ after each 4 minute dwell time? Please fix this.

174: are the concentrations of the disinfectants from the petri dishes reported anywhere, and how do they compare to what was spread?

189-210: I’m not sure it is scientifically correct to report the data as a function of time, since what you’re really doing is increasing dose or mass over time. If you had fogged the chamber one time, then taken samples after that at discrete time points, then it would be ok to report inactivation as a function of time. But you have the two variables (time and mass of disinfectant dispensed or dose) intertwined.

212: regarding this subsection, that you would get increased mass of disinfectant over cumulative time seems a bit obvious, so authors need to clarify what is new about this.

241: you should indicate here what the inactivation rates are, instead of just saying they are derived from the slopes.

253: clarify what you mean by viral solutions differed by physiological conditions

258: as mentioned before, be careful about using the word kinetics (implying rate over time), since it’s also about inactivation as a function of mass applied or dose.

268: PAA is always in equilibrium with hydrogen peroxide.

1st paragraph in Discussion: The authors need to clarify what was new/significant about their research. If no one has looked at using fogging of hydrogen peroxide or hypochlorous acid to inactivate SARS CoV 2 or influenza, then state this. If there have been studies looking at that, then you need to reference those and further differentiate your study from them.

269: if dry fogging is considered to be effective based on previous research, then explain why the current study was conducted. This sentence needs some work.

272: rather than say they won’t remain in the environment, you should discuss their reaction byproducts and how long those will remain in the environment. HP decays to water and O2, but what about HOCl?

272: again, to reiterate throughout the article, you need to differentiate between spraying and fogging. They are not the same.

277-278: do those studies compare influenza and SARS CoV-2 stability? If not, then referencing them doesn’t really help here with respect to saying they’re consistent or not with your study.

279: you need to clarify that to effectively inactivate both the flu and SARS virus using a similar contact time, you needed to use a higher concentration of disinfectant for the flu virus. You might have been able to inactivate the flu virus effectively with a lower concentration of disinfectant, but it may have required a longer contact time.

285: indicate whether you saw excessive condensation on the chamber walls when you were doing your experiments. Did opening the chamber door every 4 minutes help to remove condensation?

290: it depends on the drop size if they behave like air. They need to be really small to behave like air

294-299: you may want to clarify that fogging of disinfectant chemicals should be done with no people present in the space. If they have to be present, they should be wearing appropriate PPE and respiratory protection.

300-303: again, this is quite obvious.

316-318: this sentence makes no sense. Please rewrite.

316-328: this paragraph doesn’t really have any relevance to the study. Delete or fix.

343: I think the lipid envelope is one of the more critical physiological factors involved in viral inactivation; you may want to emphasize that more in your discussion. Lipid enveloped viruses are the least resistant microbes to disinfection: https://www.cdc.gov/infectioncontrol/guidelines/disinfection/tables/figure1.html

Figure 4: need to indicate the units for Y axis dose for A and B. Also, in Figure 4C and 4D, the X-axis says concentration, but your units say micrograms. Concentration is usually mass/volume. So need to clarify these.

497: at least 3 independent experiments? Or were they replicates? The number of replicates and/or experiments should be explained in the Materials and Methods section

General comment about figure legends: They all need to be more concise; they are too long and wordy. For example, no need to refer to and or repeat info already discussed in M and M section. If info about how the data or chart was developed or m

Reviewer #2: excellent work, well designed and thought through; see below

Line 61: correct the sentence

Line 112: specify what method used to calculate TCID50 (Reed-Muench or Spearman-Kaber)

Line 128: specify what type of BSC (Class II type A2, Class II type B2, Class III, etc.)

Show data demonstrating the effectiveness and/or interference of the neutralizers used to neutralize HAS and H2O2 on viral growth and/or tissue culture

Any thoughts on why HAS is more resistant to SARS-CoV-2 (in comparison to Influenza A) while less so against H2O2

Line 295: did you mean, “the amount of air being inhaled by the number of breaths” instead of the amount of suction…”

Line 296: consider writing “and the toxicity of the chemical to the human body”

Line 311: did you mean “the amount of disinfectant needed for the volume…”

Line 496: “n.s. not significant” are you planning to add this in the actual figure? If not, no need for this sentence here

6. PLOS authors have the option to publish the peer review history of their article (what does this mean?). If published, this will include your full peer review and any attached files.

Reviewer #1: No

Reviewer #2: No

---

## [Author Response · Author response to Decision Letter 0]

21 Feb 2022

Thank you for the opportunity to revise our manuscript, PONE-D-21-38754 entitled “Inactivation of SARS-CoV-2 and influenza A virus by spraying hypochlorous acid solution and hydrogen peroxide solution in the form of dry fog” by Urushidani, M. et al. We highly appreciate the detailed review and constructive suggestions. We consider the quality of the manuscript to have been markedly improved by the suggested edits. Changes made in the revised manuscript are marked using track changes.

We herein present the Reviewers’ comments followed by our point-by-point responses, including their locations (line numbers) in the revised manuscript. 

Reviewer #1:

In title and throughout paper, the term “dry fog” does not need to be capitalized, since it is not a proper noun. this may be an issue with English

Thank you for your comment. We corrected this term throughout the manuscript.

Line 21: you should clarify that the current thinking is that the predominant mechanism for transmission is via air (small particles, up to droplets), but that contact with contaminated surfaces may be a possibility.

According to the Reviewer’s suggestion, we revised the sentence on lines 20-22, as follows. “Severe acute respiratory syndrome coronavirus 2 (SARS-CoV-2), the causative agent of coronavirus disease 2019 (COVID-19), is transmitted mainly by droplet or aerosol infection; however, it is also transmitted by contact infection.” 

57: confusing “spraying” for “fogging” they are not the same.

Thank you for your insight. We replaced “spraying” with “fogging” or “dry fogging” throughout the text.

59-63: the sentence is a bit long, and somewhat inaccurate. Check the reference again, but after looking at it myself, it seems the CDC is ok with spraying, and using some types of foggers. Maybe the authors are confusing fogging with spraying, due to English being their second language.

Thank you for your insight. We checked the reference again, and revised the sentences on lines 61-67, as follows. “In addition, the Center for Disease Control and Prevention of the United States of America (USA) does not recommend the fogging of disinfectants in hospital rooms in the 2003 and 2008 guidelines [9]. Newer technologies for performing disinfectant fogging were assessed in the 2011 guidelines for the prevention and control of norovirus gastroenteritis outbreaks in healthcare settings; nevertheless, further research is required to clarify the effectiveness and reliability of disinfectant fogging [9].”

65: change “spraying” to “dispensing”… this comment applies throughout the paper.

Thank you for your suggestion. We changed “spraying” to “fogging” or “dry fogging” throughout the text.

67: add reference for your dry fog definition.

Thank you for your comment. We added reference #10 for the dry fog definition. 

85: indicate how pH was measured

We measured the pH of the solution using the pH meter SK-620PHⅡ (Sato Keiryoki mfg. Co., Ltd, Tokyo, Japan). We added this information to the Materials and methods section on lines 95-96, as follows. “using the pH meter SK-620PHⅡ (Sato Keiryoki Mfg. Co., Ltd, Tokyo, Japan).” 

118: suggest starting a new paragraph here in discussing inoculation of the viruses into the well plates. Also need to add some more details about this. For example, need to address how were the viruses added, how many well plates were inoculated with virus each experiment, how the positive controls were handled, etc.

Thank you for your advice. According to the Reviewer’s suggestion, we made a new paragraph “Preparation of air-dried virus samples” and added more detailed experimental methods to lines 125-132, as follows. 

“Preparation of air-dried virus samples 

Viral solutions (5 µl) containing SARS-CoV-2 (1.2 × 105 TCID50) or influenza A virus (2.8 × 106 TCID50) were applied using a micropipette to the bottom of 5 wells (per virus inactivation experiment) on a 96-well flat-bottomed microplate (Corning Japan, Shizuoka, Japan), air-dried for 10-15 minutes using a small electric fan, and subjected to a virus inactivation experiment. As a negative control for the experiment, an air-dried viral sample in a well was re-suspended with 200 µl of DMEM containing a neutralizer of the disinfectant prior to dry fogging, as described below.”

We also added more information on the handling of air-dried virus samples in virus inactivation experiments to a later paragraph (lines 146-150), as follows. “In the virus inactivation experiment using dry fogging, four 90-mm Petri dishes containing 20 ml of distilled water, a temperature and humidity sensor (Model RHT-3 Temperature and Humidity Sensor; Sensatec Co., Ltd., Kyoto, Japan), and a 96-well microplate containing air-dried viral samples in 5 wells were placed in the chamber (Fig. 1).”

130: so a sprayer was used and not a fogger? It’s confusing.

We replaced “sprayer” with “fogger” on line 141.

136: explain what the 4 petri plates of water were used for

We added an explanation on the purpose of the 4 petri dishes on lines 150-153, as follows. “Four 90-mm Petri dishes containing distilled water were placed to measure the concentrations of disinfectants trapped in water during the virus inactivation experiment and also to calculate the product of the concentrations of disinfectants and contact times (CT value).”

162: what is DPD? Confirm that the chlorine meter measures free available chlorine and not just other species of chlorine

We added an explanation for the abbreviation DPD to line 178, as follows. “DPD (N,N-dimethyl-p-phenylenediamine)” We measured free available chlorine (FAD) according to the procedure mentioned in the CDC website , https://www.cdc.gov/safewater/chlorine-residual-testing.html/. We did not measure other species of chlorine in the present study.

169: indicate what was used to measure temp and relative humidity

We added information on the equipment used to measure temperature and humidity to lines 148-149, where the procedure was described for first time in the manuscript, as follows. “a temperature and humidity sensor (Model RHT-3 Temperature and Humidity Sensor; Sensatec Co., Ltd., Kyoto, Japan)”

170-180: the paragraph about how the experiments were conducted is not very clear.

171: need to clarify how many viral samples were resuspended with the DMEM/neutralizer. I’m assuming maybe ¼ of samples were treated this way… and then another ¼ after each 4 minute dwell time? Please fix this.

Thank you for your comments. We agree with the Reviewer that the experimental procedure was not clearly described. In addition, we found some errors in the description. Therefore, we completely rewrote the experimental procedure on lines 187-213, as follows. “A set of viral inactivation experiments was performed as follows. Four 90-mm Petri dishes containing distilled water, a temperature and humidity sensor, and a 96-well microplate containing air-dried viral samples in 5 wells were placed in the test chamber (Fig. 1), as described above. Prior to the dry fogging of disinfectants, a dried viral sample in the first well was resuspended with DMEM (200 µl) containing a neutralizer by pipetting. The sliding door was then closed, and the first dry fogging of the disinfectant was performed for 5 seconds. After 4 minutes, the sliding door of the chamber was opened, the dried viral sample in the second well was resuspended with DMEM (200 µl) containing the neutralizer, and the first Petri dish containing distilled water was removed from the chamber. The sliding door was closed, and the second dry fogging of the disinfectant was performed for 2.5 seconds. After 4 minutes, the sliding door was opened, the dried viral sample in the third well was resuspended with DMEM (200 µl) containing the neutralizer, and the second Petri dish containing distilled water was removed from the chamber. The sliding door was again closed, and the third dry fogging of the disinfectant was performed for 2.5 seconds. After 4 minutes, the sliding door was opened, the dried viral sample in the fourth well was resuspended with DMEM (200 µl) containing the neutralizer, and the third Petri dish containing distilled water was removed from the chamber. The sliding door was closed, and the fourth dry fogging of the disinfectant was performed for 2.5 seconds. After 4 minutes, the sliding door was opened, the dried viral sample in the fifth well was resuspended with DMEM (200 µl) containing the neutralizer, and the fourth Petri dish containing distilled water was removed from the chamber. The concentrations of the droplet-shaped disinfectant that fell into distilled water in the Petri dishes were measured between dry fogging. Therefore, the concentrations of the disinfectant trapped in distilled water were measured 4 times in each set of experiments, i.e., at 4 minutes (total of 5 seconds of dry fogging), 8 minutes (total of 7.5 seconds of dry fogging), 12 minutes (total of 10 seconds of dry fogging), and 16 minutes (total of 12.5 seconds of dry fogging).”

174: are the concentrations of the disinfectants from the petri dishes reported anywhere, and how do they compare to what was spread?

Thank you for the comment. We added a supplementary table (S1 Table) showing raw data on disinfectant concentrations dissolved in 20 ml of distilled water in Petri dishes, and described it in the Results section on lines 253-255, as follows. “Raw data on the concentrations of disinfectants that dissolved in distilled water are shown in S1 Table.” We hope that our response to this comment is adequate.

189-210: I’m not sure it is scientifically correct to report the data as a function of time, since what you’re really doing is increasing dose or mass over time. If you had fogged the chamber one time, then taken samples after that at discrete time points, then it would be ok to report inactivation as a function of time. But you have the two variables (time and mass of disinfectant dispensed or dose) intertwined.

Thank you for your insight. We agree with the Reviewer that it is not adequate to interpret the data as “the viruses were inactivated by dry fogging of disinfectant in a time-dependent manner”. Therefore we removed the word “time-“ from lines 29 and 553 and replaced the words “in a time-dependent manner” with “over time” on lines 224 and 239. It is clear that the inactivation of viruses was observed under the experimental conditions used; therefore, we consider this modification to improve the accuracy of the interpretation of data obtained. In addition, we added sentences to explain how we conducted the experiments to lines 243-246, as follows. “It is important to note that since dry fogging was performed 4 times at 4-minute intervals in the virus inactivation experiment, the concentration of the disinfectant increased over time. Nevertheless, viruses were inactivated over time under the experimental conditions used.”

212: regarding this subsection, that you would get increased mass of disinfectant over cumulative time seems a bit obvious, so authors need to clarify what is new about this.

We agree with the Reviewer that it is obvious that the amount of exposure increased as the fogging time became longer. We described the results as “expected results” on lines 265-268. Our new result is that when we compared the dry fogged and collected (in distilled water in Petri dishes) concentrations of disinfectants, the concentration of hypochlorous acid in droplets appeared to be lower than that of hydrogen peroxide during travelling in the form of dry fog. Therefore, we added sentences to explain this in more detail to lines 270-274, as follows. “Since the fogging time was the same for hypochlorous acid solution and hydrogen peroxide solution, the number of droplets that fell and dissolved into distilled water in Petri dishes was equivalent. Therefore, the data obtained indicated that the concentration of FAC in droplets decreased more than that of hydrogen peroxide when they travelled in the form of a dry fog.”

241: you should indicate here what the inactivation rates are, instead of just saying they are derived from the slopes.

Thank you for your insight. We reconsidered how to evaluate the virucidal effects of disinfectants, and revised the sentences in the Results section on lines 278-305, as follows.

“Relationship between virucidal effects of dry fogged disinfectants and the CT value

The model of Eq ([Disp-formula pone.0261802.e001]) calculated from Chick Watson’s law is often used to show the bactericidal effects of a disinfectant on microorganisms [11].

 log (N / N0) = -kCT (1)

In the equation, N0 is the initial bacterial count, N is the viable bacterial count at the contact time (T) of bacteria to the disinfectant, C is the disinfectant concentration, and k is the inactivation rate constant of the bacteria. In the present study, we evaluated the virucidal effects of dry fogged disinfectants by CT values, the product of disinfectant concentrations and contact times, similar to the bacteria inactivation experiment. The concentration of the disinfectant trapped in 20 ml of distilled water in Petri dishes was used as the concentration (C). Figure 5 shows the logarithmic values of viral infectious titers at various CT values. The viral titers of SARS-CoV-2 and influenza A virus linearly decreased with increases in CT values in the dry fogging of hypochlorite acid solution. With the dry fogging of hydrogen peroxide solution, viral titers also linearly decreased. It is important to note that since the logarithmic values of viral infectious titers at various CT values were plotted and fit by a simple linear regression analysis, the solid lines in Figure 5 are shown by Eq ([Disp-formula pone.0261802.e002]), which is a modification of Eq ([Disp-formula pone.0261802.e001]).

In the equation, I0 is the viral titer before dry fogging, I is the viral titer at the contact time (T) of the virus to a dry fogged disinfectant, and C is the concentration of a disinfectant trapped in 20 ml of distilled water. 

In formula ([Disp-formula pone.0261802.e002]), k is the inactivation rate constant of the virus. With the dry fogging of hypochlorous acid solution, the inactivation constant of SARS-CoV-2 was approximately 46-fold smaller than that of influenza A virus. In addition, with the dry fogging of hydrogen peroxide solution, the inactivation constant of SARS-CoV-2 was approximately 8.8-fold smaller than that of influenza A virus. Therefore, SARS-CoV-2 was more resistant to hypochlorous acid solution and hydrogen peroxide solution than influenza A virus.”

We decided to evaluate the viricidal effects of dry fogged disinfectants by comparing viral infectious titers at various CT values calculated by the concentrations of disinfectants trapped in 20 ml of distilled water in Petri dishes and contact times. Accordingly, we replaced the term “exposed disinfectant amount” with “concentration of a disinfectant” or an equivalent throughout the text. In addition, we revised Figures 4 and 5 as well as the figure legends. We hope the Reviewer agrees with the revised method for evaluating the virucidal effects of dry fogged disinfectants.

253: clarify what you mean by viral solutions differed by physiological conditions

We modified the sentence on lines 310-312, as follows. “We prepared air-dried viral samples using viral supernatants for these experiments, and viral solutions differed from body fluids.”

258: as mentioned before, be careful about using the word kinetics (implying rate over time), since it’s also about inactivation as a function of mass applied or dose.

Thank you for your comments. We modified the sentence on lines 316-318, as follows. “The results obtained revealed no significant differences in the levels of virus inactivation upon the dry fogging of disinfectants between air-dried viral samples prepared with artificial saliva and PBS (Fig. 6).”

268: PAA is always in equilibrium with hydrogen peroxide.

Thank you for your insight. We corrected the sentence on lines 326-327, as follows. “the dry fogging of a mixture of peracetic acid and hydrogen peroxide was previously demonstrated [16-21].”

1st paragraph in Discussion: The authors need to clarify what was new/significant about their research. If no one has looked at using fogging of hydrogen peroxide or hypochlorous acid to inactivate SARS CoV 2 or influenza, then state this. If there have been studies looking at that, then you need to reference those and further differentiate your study from them.

269: if dry fogging is considered to be effective based on previous research, then explain why the current study was conducted. This sentence needs some work.

Thank you for your comment. We consider this to be the first study to demonstrate the inactivation of SARS-CoV-2 by the dry fogging of hypochlorous acid solution or hydrogen peroxide alone. Therefore, we added this information to lines 334-336, as follows. “To the best of our knowledge, the inactivation of SARS-CoV-2 by the dry fogging of hypochlorous acid solution or hydrogen peroxide has not yet been reported.”

272: rather than say they won’t remain in the environment, you should discuss their reaction byproducts and how long those will remain in the environment. HP decays to water and O2, but what about HOCl?

Thank you for your insight. We modified the sentences on lines 329-334, as follows. “Hypochlorous acid solution and hydrogen peroxide solution were employed as disinfectants to be dry fogged in the present study. Hypochlorous acid is decomposed into hydrochloric acid and oxygen, while hydrogen peroxide is decomposed into water and oxygen with time. Since decomposition products other than water are gases at a normal temperature and pressure, they do not remain on environmental surfaces after dry fogging with appropriate ventilation.”

272: again, to reiterate throughout the article, you need to differentiate between spraying and fogging. They are not the same.

Thank you for the comment. We replaced “spraying” with “fogging” or “dry fogging” throughout the text.

277-278: do those studies compare influenza and SARS CoV-2 stability? If not, then referencing them doesn’t really help here with respect to saying they’re consistent or not with your study.

279: you need to clarify that to effectively inactivate both the flu and SARS virus using a similar contact time, you needed to use a higher concentration of disinfectant for the flu virus. You might have been able to inactivate the flu virus effectively with a lower concentration of disinfectant, but it may have required a longer contact time.

Thank you for your insight. We agree that the studies cited in references 4, 5 and 7 did not compare the stabilities of influenza A virus and SARS-CoV-2. In addition, although the study cited in reference 6 compared their stabilities on human skin as well as on environmental surfaces, their findings and the present results were not similar. Furthermore, our description of fogging was insufficient. Therefore, we modified the sentences on lines 340-344, as follows. “A previous study reported the higher stability of SARS-CoV-2 than influenza A virus on environmental surfaces [6]. We speculated that the higher stability of SARS-CoV-2 than influenza A virus may be related to our results showing that with dry fogging for the same duration, higher concentrations of hypochlorous acid solution and hydrogen peroxide solution were required to inactivate SARS-CoV-2 than influenza A virus.”

285: indicate whether you saw excessive condensation on the chamber walls when you were doing your experiments. Did opening the chamber door every 4 minutes help to remove condensation?

Thank you for your comment. We did not see excessive condensation on the chamber walls under the experimental conditions used. However, the floor of the chamber was a little wet after the experiments. We attributed this to the falling of dry fog to the floor. We do not know the extent to which opening of the chamber door affected condensation.

290: it depends on the drop size if they behave like air. They need to be really small to behave like air

Thank you for your insight. We remove the words “behave like air” and modified the sentence on lines 355-356, as follows. “Furthermore, droplets reach the backs of objects, such as desks and chairs, as well as gaps that cannot be accessed;”

294-299: you may want to clarify that fogging of disinfectant chemicals should be done with no people present in the space. If they have to be present, they should be wearing appropriate PPE and respiratory protection.

Thank you for your insight. We added the sentence on lines 364-365, as follows. “Therefore, if someone has to be present, they need to be wearing appropriate personal protective equipment and respiratory protection.” 

300-303: again, this is quite obvious.

Thank you for the comment. We wish to keep sentences because we would like to show the context to the general readers.

316-318: this sentence makes no sense. Please rewrite.

Thank you for the comment. We rewrote the sentences on lines 382-384, as follows. “Three important factors for the occurrence of infectious diseases are infection sources, transmission routes, and susceptible hosts; therefore, countermeasures need to be taken against these factors.”

316-328: this paragraph doesn’t really have any relevance to the study. Delete or fix.

Thank you for your insight. Although the Reviewer considers the paragraph to be unnecessary, we would like to explain 3 important factors involved in the occurrence of infectious diseases to the general readers. In addition, we would like to mention that the inactivation of pathogens on environmental surfaces is a countermeasure against an infection source. Therefore, please let us keep this paragraph. We hope that the Reviewer agrees with our opinion.

343: I think the lipid envelope is one of the more critical physiological factors involved in viral inactivation; you may want to emphasize that more in your discussion. Lipid enveloped viruses are the least resistant microbes to disinfection: https://www.cdc.gov/infectioncontrol/guidelines/disinfection/tables/figure1.html

Thank you for your insight. We agree with you, and added a sentence to lines 410-411 in order to emphasize it, as follows. “A lipid bilayer is the least resistant component of enveloped viruses to disinfection [9];”

Figure 4: need to indicate the units for Y axis dose for A and B. Also, in Figure 4C and 4D, the X-axis says concentration, but your units say micrograms. Concentration is usually mass/volume. So need to clarify these.

Thank you for pointing out the lack of an explanation in Fig. 4. We revised the Figure accordingly.

497: at least 3 independent experiments? Or were they replicates? The number of replicates and/or experiments should be explained in the Materials and Methods section

Thank you for the suggestion. We added an explanation to the Materials and methods section on lines 213-215, as follows. “Virus inactivation experiments under the same conditions were repeated more than three times, except for those to evaluate artificial saliva, which were repeated twice.” Since most of the viral inactivation experiments conducted under the same conditions were performed more than three times, please let us keep the explanation in the Figure legends.

General comment about figure legends: They all need to be more concise; they are too long and wordy. For example, no need to refer to and or repeat info already discussed in M and M section. If info about how the data or chart was developed or m

Thank you for your comment. We shortened the figure legends. We hope the general readers understand the context of the figures easily without repeatedly reading the main text. We hope the Reviewer agrees with our opinion.

Reviewer #2:

excellent work, well designed and thought through; see below

Thank you for your comments.

Line 61: correct the sentence

We rewrote the sentences on lines 61-67 as to be more accurate, as follows. “In addition, the Center for Disease Control and Prevention of the United States of America (USA) does not recommend the fogging of disinfectants in hospital rooms in the 2003 and 2008 guidelines [9]. Newer technologies for performing disinfectant fogging were assessed in the 2011 guidelines for the prevention and control of norovirus gastroenteritis outbreaks in healthcare settings; nevertheless, further research is required to clarify the effectiveness and reliability of disinfectant fogging [9].”

Line 112: specify what method used to calculate TCID50 (Reed-Muench or Spearman-Kaber)

Thank you for pointing out the lack of an explanation. We used the Spearman-Kaber method. The sentence on lines 116-119 was revised, as follows. “median tissue culture infectious doses (TCID50) were measured using the Spearman-Kaber method following the fixation of cells with 5% formaldehyde in phosphate-buffered saline (PBS) and staining with 0.5% crystal violet in 20% EtOH.”

Line 128: specify what type of BSC (Class II type A2, Class II type B2, Class III, etc.)

Thank you for your comment. We used a biosafety cabinet class II type A/B3 in the P3 room, while a biosafety cabinet class II type A1 in the P2 room. Therefore, the sentence on lines 139-140 was modified, as follows. “set in a biosafety cabinet of class II type A/B3 or class II type A1.” 

Show data demonstrating the effectiveness and/or interference of the neutralizers used to neutralize HAS and H2O2 on viral growth and/or tissue culture

Thank you for your comment. These neutralizers did not affect cell growth, at least under our experimental conditions; therefore, we added the sentence “These neutralizers did not affect cell growth under experimental conditions (data not shown).” to lines 171-172. The effects of the neutralizers on viral growth were not examined; however, since we included the reagents in all virus inactivation experiments, including a negative control one, we consider the effects on viral growth to not have affected the experimental outcomes. We hope the Reviewer agrees with our opinion.

Any thoughts on why HAS is more resistant to SARS-CoV-2 (in comparison to Influenza A) while less so against H2O2

Thank you for your comment. We consider the different stabilities of those viruses to reflect their susceptibilities to the disinfectants examined, as mentioned in the Discussion section, lines 340-344, as follows. “A previous study reported the higher stability of SARS-CoV-2 than influenza A virus on environmental surfaces [6]. We speculated that the higher stability of SARS-CoV-2 than influenza A virus may be related to our results showing that with dry fogging for the same duration, higher concentrations of hypochlorous acid solution and hydrogen peroxide solution were required to inactivate SARS-CoV-2 than influenza A virus.” However, we were unable to elucidate this in more detail in the present study.

Line 295: did you mean, “the amount of air being inhaled by the number of breaths” instead of the amount of suction…”

Line 296: consider writing “and the toxicity of the chemical to the human body”

Thank you for your comments. According to your suggestion, we revised the sentences on lines 358-361, as follows. “Therefore, several factors need to be considered, such as the concentration of droplets in space, the staying time, the amount of air being inhaled by the number of breaths, the concentration of the solution in droplets, and the toxicity of the chemical to the human body.”

Line 311: did you mean “the amount of disinfectant needed for the volume…”

Thank you for pointing out the unclear description. We revised the sentence on lines 376-377, as follows. “it is inadequate to simply calculate the amount of disinfectant needed for the volume of the space to be fogged.”

Line 496: “n.s. not significant” are you planning to add this in the actual figure? If not, no need for this sentence here

Thank you for your comment. Since Fig. 3A contains “not significant” results, we would like to keep it.

---

## [Decision Letter · Decision Letter 1]

9 Mar 2022

PONE-D-21-38754R1Inactivation of SARS-CoV-2 and influenza A virus by dry fogging hypochlorous acid solution and hydrogen peroxide solutionPLOS ONE

Dear Dr. Kameoka,

Thank you for submitting your manuscript to PLOS ONE. After careful consideration, we feel that it has merit but does not fully meet PLOS ONE’s publication criteria as it currently stands. Therefore, we invite you to submit a revised version of the manuscript that addresses the points raised during the review process. The comments raised by the reviewer seem minor.

We look forward to receiving your revised manuscript.

Kind regards,

Etsuro Ito

Academic Editor

PLOS ONE

Journal Requirements:

Reviewers' comments:

Reviewer's Responses to Questions

**Comments to the Author**

1. If the authors have adequately addressed your comments raised in a previous round of review and you feel that this manuscript is now acceptable for publication, you may indicate that here to bypass the “Comments to the Author” section, enter your conflict of interest statement in the “Confidential to Editor” section, and submit your "Accept" recommendation.

Reviewer #1: (No Response)

2. Is the manuscript technically sound, and do the data support the conclusions?

Reviewer #1: Yes

3. Has the statistical analysis been performed appropriately and rigorously? 

Reviewer #1: Yes

4. Have the authors made all data underlying the findings in their manuscript fully available?

Reviewer #1: Yes

5. Is the manuscript presented in an intelligible fashion and written in standard English?

Reviewer #1: Yes

6. Review Comments to the Author

Reviewer #1: I think you still need to make the figure captions more concise. A lot of the verbiage in the captions I believe is already described in the materials/methods section, or described in the Results section.

I think the Discussion section could be made more concise as well, especially the last two paragraphs. this seems like a repeat of the Introduction.

In the abstract and elsewhere, where you say that it is also transmitted by contact infection, i think maybe you should say it may be transmitted by surface contact. Use the word "may".

7. PLOS authors have the option to publish the peer review history of their article (what does this mean?). If published, this will include your full peer review and any attached files.

Reviewer #1: No

---

## [Author Response · Author response to Decision Letter 1]

22 Mar 2022

Thank you for the opportunity to revise our manuscript, PONE-D-21-38754R1 entitled “Inactivation of SARS-CoV-2 and influenza A virus by dry fogging hypochlorous acid solution and hydrogen peroxide solution” by Urushidani, M. et al. We appreciate the detailed review and suggestions made by the reviewer #1. We herein present the Reviewers’ comments, followed by our point-by-point responses. Changes made in the revised manuscript are marked using track changes. 

Reviewer #1: I think you still need to make the figure captions more concise. A lot of the verbiage in the captions I believe is already described in the materials/methods section, or described in the Results section.

(Response) Thank you for your suggestion. We shortened the figure legends.

I think the Discussion section could be made more concise as well, especially the last two paragraphs. this seems like a repeat of the Introduction.

(Response) Thank you for your suggestion. We removed some sentences that overlapped with the Introduction section, and made the Discussion section more concise. Please let us remain some sentences “overlapped” with the Results section, because we would like to emphasize the results. We hope the reviewer agrees with our opinion.

In the abstract and elsewhere, where you say that it is also transmitted by contact infection, i think maybe you should say it may be transmitted by surface contact. Use the word "may". 

(Response) Thank you for the suggestion. We added “may” in the corresponding sentence in the Abstract section.

---

## [Editor Report · Decision Letter 2]

24 Mar 2022

Inactivation of SARS-CoV-2 and influenza A virus by dry fogging hypochlorous acid solution and hydrogen peroxide solution

PONE-D-21-38754R2

Dear Dr. Kameoka,

We’re pleased to inform you that your manuscript has been judged scientifically suitable for publication and will be formally accepted for publication once it meets all outstanding technical requirements.

Kind regards,

Etsuro Ito

Academic Editor

PLOS ONE

---

## [Editor Report · Acceptance letter]

29 Mar 2022

PONE-D-21-38754R2 

Inactivation of SARS-CoV-2 and influenza A virus by dry fogging hypochlorous acid solution and hydrogen peroxide solution 

Dear Dr. Kameoka:

I'm pleased to inform you that your manuscript has been deemed suitable for publication in PLOS ONE. Congratulations! Your manuscript is now with our production department. 

Kind regards, 

on behalf of

Prof. Etsuro Ito 

Academic Editor

PLOS ONE